# Preoperative Anxiety Impact on Anesthetic and Analgesic Use

**DOI:** 10.3390/medicina59122069

**Published:** 2023-11-23

**Authors:** Hanaa Baagil, Hamzah Baagil, Mark Ulrich Gerbershagen

**Affiliations:** 1Department of Anaesthesiology, Hospital Cologne Holweide, Teaching Hospital of the University Cologne, Neufelder Str. 32, 51067 Cologne, Germany; 2Department of Neurology, RWTH Aachen University, Pauwelsstraße 30, 52074 Aachen, Germany; hbaagil@ukaachen.de; 3JARA-BRAIN Institute Molecular Neuroscience and Neuroimaging, Research Center Jülich, RWTH Aachen University, 52074 Aachen, Germany

**Keywords:** preoperative anxiety, postoperative pain, anesthetic use, pain perception, anxiety on anesthesia and surgery

## Abstract

Anxiety is a complex emotional state that can arise from the anticipation of a threatening event, and preoperative anxiety is a common experience among adult patients undergoing surgery. In adult patients, the incidence of preoperative anxiety varies widely across different surgical groups, and it can result in a variety of psychophysiological responses and problems. Despite its negative impact, preoperative anxiety often receives insufficient attention in clinical practice. To improve pain management strategies, there is a need for further research on personalized approaches that take into account various factors that contribute to an individual’s pain experience. These personalized approaches could involve developing tools to identify individuals who are more likely to experience increased pain and may require additional analgesia. To address this, regular assessments of anxiety levels should be conducted during preoperative visits, and counseling should be provided to patients with high levels of anxiety. Identifying and addressing preoperative anxiety in a timely manner can help reduce its incidence and potential consequences.

## 1. Introduction

Anxiety represents a multifaceted emotional condition that can appear as either fear or the anticipation of an imminent danger, and it can lead to various manifestations encompassing behavioral, emotional, cognitive, and physical indicators. The perioperative period is often regarded as one of the most unsettling phases for surgical patients, with anxiety being frequently cited as the most daunting aspect of the preoperative stage. The prevalence of preoperative anxiety is high and carries considerable consequences for how anesthesia is administered and, ultimately, the overall outcomes of surgical procedures [1,2,3,4]. While a certain level of anxiety is expected during the preoperative phase, excessive fear and multi-systemic manifestations can become a clinical problem.

Preoperative anxiety is known to increase anesthetic agent requirement, delay awakening, and cause hemodynamic derangements, postoperative pain, wound healing delay, infection risk, cancellation, a longer hospital stay, and dissatisfaction [5,6,7,8]. It is believed that preoperative anxiety originates from the activation of the autonomic nervous system, which subsequently triggers neuroendocrine alterations, ultimately leading to an elevated heart rate, increased blood pressure, and myocardial workload. Furthermore, the presence of preoperative anxiety has been recognized as one of the contributors to long-term morbidity and mortality after cardiac surgery [9,10,11,12,13]. The incidence of preoperative anxiety varies based on several factors, such as the nature of the surgical procedure, gender, the criteria applied, the reason for the surgery, and level of education [1,2,3,4,14,15,16].

It remains unclear whether preoperative anxiety or postoperative pain exerts a greater influence on the occurrence of postoperative delirium. Research findings have indicated that there was no significant association between preoperative anxiety and postoperative delirium among elderly patients [9,14,17,18]. While the relationship between preoperative anxiety, postoperative pain, and postoperative delirium is complex, it is important to clarify these connections. Understanding how preoperative anxiety, postoperative pain, and postoperative delirium are connected is crucial for the investigation into the impact of preoperative anxiety on pain relief and the occurrence of postoperative delirium. For example, a study conducted by Saho Wada et al. [14,19] found that preoperative anxiety is a predictor of postoperative delirium in people undergoing cancer-related surgeries. Delirium, a significant cognitive disorder, often occurs after major elective surgeries in older patients, with an incidence of 15 to 25% [14,20]. Furthermore, understanding the link between preoperative anxiety, postoperative pain, and postoperative delirium is vital for exploring how high postoperative pain and the use of opioids can increase the risk of postoperative delirium and how effective pain management can reduce this risk. Notably, a study by Liu Q et al. [14] suggested that high postoperative pain might predict postoperative delirium in elderly patients undergoing gastrointestinal cancer surgery, although the exact mechanism is still unclear and needs further investigation. In summary, recognizing the connection between preoperative anxiety, postoperative pain, and postoperative delirium is essential as it highlights the various factors that can affect patient outcomes in the perioperative setting, especially concerning pain relief and preventing postoperative delirium.

Preoperative anxiety has been demonstrated to have a crucial influence on postoperative pain modulation [14,21,22,23]. Existing studies and research have centered on the concept of a linear, not curvilinear connection between anxiety and pain, signifying that heightened anxiety corresponds to increased pain levels [14,24]. Prior psychological interventions targeting preoperative anxiety in patients have been proven to be effective in diminishing the severity of postoperative pain [14,25]. However, Kain et al. [1,13] highlighted that in the context of patients undergoing major surgery, there may not be a significant positive correlation between anxiety and pain, suggesting that the usual association between pain and anxiety might not apply in this specific patient population. A study conducted by Kalkman et al. [1,14] similarly determined that preoperative pain, rather than preoperative anxiety, emerged as the most robust predictor of severe postoperative pain.

Pain is a multifaceted subjective phenomenon that includes various aspects, such as sensory discriminative, emotional–affective, and cognitive evaluative elements. The intensity of postoperative pain is frequently linked to intense nociceptive stimulation from traumatized tissue, inadequate analgesia, and the emotional state of the patient, which can influence their behavioral response and anxiety levels. Understanding the association between preoperative anxiety and postoperative pain can help mitigate their synergetic effect.

Pharmacological medications have been commonly used to manage anxiety in medical settings, but they often come with unwanted side effects. Alternatively, non-pharmacological approaches like providing reassurance, music therapy, practicing breathing exercises, meditation, using acupressure, and offering pre-procedure education have demonstrated their effectiveness in reducing preoperative anxiety. These interventions are not only cost-effective but also easy to perform, without requiring a high level of technical skill or equipment, and most importantly, are free of adverse effects. By incorporating non-pharmacological interventions into preoperative care, healthcare providers can provide a more comprehensive and patient-centered approach to reduce preoperative anxiety, ultimately improving patient outcomes and satisfaction.

## 2. Materials and Methods

The primary objective of this narrative literature review was to systematically collect and summarize existing data on the impact of preoperative anxiety on anesthetic and analgesic use in surgical adult patients in a comprehensive manner. The databases of PubMed, Cochrane Library, Medline, and Google Scholar were searched for the relevant articles from January 2013 until August 2023. The searching terms included “preoperative anxiety”, “anxiety”, “postoperative pain”, “anxiety of anesthesia and surgery”. Only studies in the literature with patients who underwent surgery or interventions in general anesthesia, regional anesthesia or analgosedation were included. Articles that provided no abstract were excluded. The inclusion criteria were established based on the primary outcome of heightened postoperative pain potentially linked to preoperative anxiety, aiming to encompass a broad and impartial selection of pertinent research studies.

## 3. Results

### 3.1. Risk Factors for Preoperative Anxiety

Preoperative anxiety is a prevalent condition, with its occurrence ranging from 11% to 80%, dependent on the assessment methodology employed [15,16]. Anxiety represents an emotional turmoil that can be attributed to either an individual’s inherent personality traits (known as trait anxiety) or the specific circumstances they are facing (referred to as state anxiety). It is characterized by feelings of distress and fear, and activation of the autonomic nervous system, and has been linked to similar psychopathological reactions as those observed in cases of acute pain [17]. Prior research has indicated that anxiety might enhance the perception of pain as individuals tend to become more vigilant and attentive to their pain sensations [26,27,28]. Conversely, postoperative pain may contribute to a cycle of pain and anxiety by stimulating a heightened anxiety response [26]. Nevertheless, conflicting findings have emerged in the literature, as certain studies have documented divergent outcomes; one set of studies has shown that reducing preoperative anxiety results in a reduction in postoperative pain, while another set of studies has observed no alteration in the postoperative pain response [26,29,30]. The variability in findings may be due to the lack of homogeneity in the populations studied with respect to gender, medical and psychological history, indication for the surgical procedure, and anesthetic management [31]. Hence, only a comprehensive, large-scale study capable of mitigating the influence of these confounding factors will have the capacity to provide a conclusive answer to this question [31].

Age has been found to be significantly correlated with the intensity of postoperative pain, with elderly patients typically requiring less postoperative analgesia than younger patients undergoing similar procedures [32]. Nevertheless, it is essential to exercise caution when interpreting these findings, as evaluations of pain and requests for pain management may be affected by confounding factors such as cognitive functioning and physical capabilities [31].

Pain is a complex multidimensional experience, shaped by a combination of psychosocial and biological factors, and as a result, gender can play a role in shaping individual variations in pain perception and pain management [17]. Many studies have identified noteworthy gender disparities in acute pain, including differences in pain thresholds, perception, tolerance, coping strategies, and the necessity for analgesic interventions [33,34,35]. Specifically, women tend to report more negative pain responses than men, suggesting that gender differences do indeed exist in pain perception [36,37].

In their study, Maheshwari et al. examined preoperative anxiety levels in patients choosing between general or regional anesthesia for elective cesarean section procedures. They found that the overall prevalence of anxiety was quite high, affecting 72.7% of the patients (112 out of 154). Interestingly, the anxiety rate was significantly elevated among patients opting for general anesthesia compared to those choosing regional anesthesia (97.2% [69 out of 71] vs. 51.8% [43 out of 83]; *p* < 0.01). Furthermore, the study identified several statistically significant associations between preoperative anxiety (defined as VAS ≥ 50) and specific factors, including an age of younger than 25 years, employment status as a working woman, nulliparous and primiparous status, lack of previous anesthesia experience, prior anesthesia experience under general anesthesia, and obtaining information from sources other than anesthetists. As a result, the authors concluded that anxiety played a significant role in patients’ decisions to decline regional anesthesia, highlighting the importance of routinely assessing anxiety in all elective cesarean section patients during preoperative anesthesia evaluations. Implementing this assessment could potentially mitigate anxiety levels and aid patients in making well-informed choices regarding their preferred anesthesia technique.

While age and female gender have consistently emerged as distinct predictors of preoperative anxiety in numerous research investigations [28,29,30], a singular, definitive factor contributing to preoperative anxiety remains elusive. Further research endeavors are essential to shed light on the specific patient attributes and surgical variables responsible for the onset of preoperative anxiety. This comprehensive understanding will pave the way for tailoring assessment and treatment approaches to address preoperative anxiety within specific patient subgroups, ultimately reducing its occurrence.

### 3.2. Measurement of Preoperative Anxiety

The assessment of preoperative anxiety can be accomplished using either objective or subjective methods [31]. Objective approaches encompass both indirect and direct evaluations of anxiety levels, involving the estimation of sympathoadrenal system activation and the measurement of stress hormones, respectively [28,31]. Numerous studies have demonstrated that elevated preoperative anxiety levels are associated with increased heart rate, blood pressure, plasma cortisol, and catecholamine levels, alongside a reduction in heart rate variability and oxygen saturation [28,32]. For instance, a study conducted by Balasubramaniyan N. et al. in 2016 revealed that hypertensive patients undergoing dental procedures experienced significant elevations in heart rate and systolic blood pressure due to preoperative anxiety. Additionally, recent research suggests that measuring catechol-o-methyltransferase levels may serve as an indicator of preoperative anxiety, as anxious individuals tend to exhibit lower levels of this enzyme and higher levels of circulating catecholamines during the preoperative period [33]. Nonetheless, it is important to note that objective techniques for assessing anxiety are time-consuming and may not be easily implemented in a busy clinical practice. In addition, the availability of several reliable and user-friendly subjective scales has further reduced the use of objective anxiety measurements.

Subjective scales are the most commonly used method for assessing preoperative anxiety. Several reliable scales are available, such as the Hospital Anxiety and Depression scale (HADS) [38], Visual Analogue Scale for Anxiety (VAS-A) [39,40], State-Trait Anxiety Inventory (STAI) [41], Amsterdam Preoperative Anxiety and Information Scale (APAIS) [42], Linear Analog Anxiety Scale (LAAS), and Multiple Affect Adjective Check List (MAACL) [43]. The VAS-A is the simplest to use, consisting of a 100 mm horizontal line, marked with zero at its left end and 100 at its right end. The main disadvantage of this scale is that it does not have a precisely established cut-off point. However, a score over 46 mm represents a clinically significant level of anxiety, while a score of ≥70 mm correlates with very high levels of anxiety [10,44]. The APAIS scale [42] is a widely accepted instrument for measuring preoperative anxiety. Developed in 1996 by Moerman et al., the APAIS scale consists of six questions grouped into two components: the first one measures anxiety related to anesthesia and surgery (four questions), and the second assesses the need for information (two remaining questions). The questions are scored from 1 to 5 based on the Likert method ranging from “not at all” to “extremely”, respectively. The total score ranges from 4 to 20 points for the anxiety component and from 2 to 10 points for the information component. A higher score indicates a higher level of anxiety and a greater need for information. However, the main limitation of the APAIS scale is its inability to distinguish anxiety related to anesthesia from anxiety related to surgery [42].

The State-Trait Anxiety Inventory (STAI) [41] is widely regarded as a gold standard tool due to its established validity and reliability, consistently producing reliable results across various populations and ethnic groups when measuring anxiety [19,20]. Consequently, many studies have opted for the STAI as their preferred instrument for evaluating preoperative anxiety. This choice is motivated by the desire to steer clear of subjective assessments, which may inadvertently lead to an overestimation of patients’ anxiety levels [45].

Choosing the most suitable scale for gauging preoperative anxiety should be influenced by various factors, including the available assessment time, the patients’ individual attributes and existing medical conditions, the preferences of the healthcare provider, and the proven reliability of a particular assessment tool. Nevertheless, it remains crucial to emphasize that the ultimate scale choice is not as critical as the timely identification of anxious patients and the implementation of interventions to alleviate preoperative anxiety [28]. Measuring preoperative anxiety is essential for providing adequate patient care, and both objective and subjective approaches should be used based on the availability of resources and clinical setting.

### 3.3. Management of Preoperative Anxiety

Preoperative anxiety is a well-known issue that affects many patients, but despite extensive research, the most effective way to manage it remains a topic of debate. Adding to the complexity of the matter is the absence of clear guidelines for preventing or treating patients with preoperative anxiety [46]. As such, there is a pressing need for further research and guidance in this area to improve patient outcomes and ensure the best possible care.

Currently, a range of strategies and interventions are available to mitigate the prevalence of preoperative anxiety and alleviate its associated symptoms [28]. Foremost among these is patient education, which stands as the primary and widely employed approach. While numerous studies have demonstrated its effectiveness [38,39,40], questions persist regarding the optimal methods for delivering patient education. Some experts advocate for the use of phone, written materials, or video resources [41,42], while other research suggests that direct personal interaction and verbal communication significantly alleviate anxiety symptoms [43,44]. Additionally, modern technologies, including mobile applications [46], online resources [47], and even virtual reality tools [48], have proven effective in managing anxious patients.

However, there are instances where educational interventions may not yield desired outcomes, be it owing to patient-specific traits, constraints on time, or the intricate nature of the underlying causes of anxiety [28]. In these scenarios, a more traditional pharmacological approach may prove valuable, encompassing options like benzodiazepines, sympatholytics, gabapentinoids, and antidepressants [28]. Notably, recent research has indicated that even melatonin can exert a beneficial impact in diminishing the extent of preoperative anxiety [16]. Nonetheless, it is important to note that some authors refute the anxiolytic effects of frequently used medications, and pharmacotherapy has its limitations and can increase treatment costs. Therefore, alternative and less expensive methods for managing anxious patients have gained significant interest in recent years [43]. These include aromatherapy [47,48], music therapy [49], acupuncture [50], and even therapeutic inhaled essential oils [51].

While research on the precise mechanisms responsible for the therapeutic and anxiety-reducing effects of music remains relatively scarce, available evidence points to the possibility of physiological changes being triggered by music through connections between brain regions within the limbic system and the functioning of endocrine and autonomic mechanisms [52,53]. Music has been found to suppress the activation of the sympathetic nervous system, resulting in a reduction in adrenergic activity [54,55]. In addition, music interventions may alleviate negative emotions by providing distraction, enhancing mood, and invoking positive memories [52,53]. Encouragingly, a recent study revealed that a music intervention led to a significant decrease in heart rate and blood pressure [52,54]. However, the clinical relevance of these findings remains uncertain, and additional research is necessary to further explore this issue. Accordingly, a systematic review emphasized the importance of high-quality studies that provide adequate information on the music intervention applied to improve the clinical significance of music interventions [54,56].

Despite these options, at present, there is a dearth of studies that have undertaken a comprehensive comparison of all available interventions, leaving the most efficacious approach for alleviating preoperative anxiety still undetermined.

To summarize, preoperative anxiety remains a persistent concern for healthcare practitioners, and the quest for the most effective management strategy continues. Until additional research is undertaken, clinical assessment remains paramount, with interventions to be customized to suit each patient’s unique requirements.

### 3.4. Relevance of Preoperative Anxiety in Clinical Practice

A great value of evidence has demonstrated that preoperative anxiety has a substantial impact on the outcomes of perioperative patients, especially those at high risk, such as individuals with cardiac disease, hypertension, diabetes mellitus, advanced age, pre-existing psychological conditions, or heightened susceptibility to anxiety [57,58,59,60,61]. Despite this knowledge, preoperative anxiety assessment is not consistently integrated into the standard preoperative evaluation and preparation for surgical procedures. As a result, it is imperative to incorporate preoperative anxiety screening, patient education on anxiety reduction strategies, raising awareness about anesthesia and surgery, and presenting postoperative pain management options as integral components of the preoperative patient assessment and preparation process.

## 4. Discussion

The perioperative period is critical for patients, and anxiety is a common psychological reaction before surgery, particularly as the date approaches, when symptoms can worsen [14]. Fear of surgery and anesthesia, surgical complications, and discomfort and pain during or after surgery are all predispositions for anxiety in surgical patients. Mild anxiety prior to surgery is a normal psychological response and is not harmful to patients. However, when anxiety levels are high, it can lead to several negative effects, such as an increase in the amount of anesthetic drugs required during surgery, worsening of postoperative pain, suppression of the immune system, and delays in the healing process [62,63,64]. It has been shown that high anxiety before surgery is not only moderately associated with higher levels of postoperative pain, but it is also considered an independent risk factor for it [14].

Patient communication and education are pivotal factors in addressing preoperative anxiety. Research has demonstrated that the capability of healthcare providers to address patient inquiries not only effectively reduces preoperative anxiety but also enhances overall patient satisfaction [28]. Overcoming the challenges of effective preoperative communication may necessitate a multi-dimensional approach that can enhance patient comprehension and satisfaction levels.

In essence, heightened levels of anxiety have been consistently linked to a reduced pain tolerance, and increased reliance on analgesics and sedation, thereby impeding post-therapeutic recovery and prolonging hospital stays [1,62,63]. Music holds a special place in human existence, with its unique elements of melody, sound, harmony, rhythm, texture, structure, and expression, all perceived uniquely by individuals [64,65,66]. Exposure to pleasant auditory stimuli has a discernible impact on hormone expression, such as cortisol, and modulates the autonomic nervous system by diminishing sympathetic activity while simultaneously enhancing parasympathetic nerve function [64,67,68]. However, its influence transcends the physical realm, encompassing pain intensity and stress response and extending to the emotional realm.

Concerning peri-procedural outcomes, three distinct effects are currently under consideration: emotional comfort, distraction from pain, and the ability to counteract feelings of vulnerability and loss of control by serving as a familiar stimulus in an unfamiliar hospital environment [64,69,70]. Consequently, music has the potential to alleviate anxiety, and since anxiety itself is known to impact the perception of pain [64,69,71], this interplay can result in a reduced need for sedation and analgesics during medical procedures [64,71,72,73,74]. The existing body of knowledge regarding the overarching impact of music on analgesia is derived from neuroimaging studies, which indicate that reduced pain sensitivity is associated with alterations in endogenous opioid levels [64,65,71,75].

Studies have consistently shown that preoperative anxiety has a positive correlation with postoperative pain [23,65,66]. High levels of preoperative anxiety can lead to poor postoperative pain control and increased morbidity, making it an important predictor of postoperative pain [67,68,69]. One possible mechanism of preoperative anxiety is that it diminishes presynaptic GABA release and inhibits the action of postsynaptic GABA receptors, which participate in the regulation of pathological pain and inhibit hyperalgesia [70,71].

Anxiety in the context of “catastrophic pain”, i.e., a strong negative reaction to real or expected pain, is a critical factor in postoperative pain management, as it has been shown to have a strong negative impact on patients’ outcomes [72]. Previous studies have established a clear link between preoperative anxiety and increased postoperative pain, with higher levels of anxiety correlating with greater postoperative analgesic requirements [73]. This emphasizes the need for anesthesiologists and surgeons to prioritize the assessment and management of preoperative anxiety in their patients, as the timely adjustment of perioperative medication is crucial for achieving optimal pain management and individualized care. Therefore, addressing preoperative adverse emotions and providing tailored care is essential for ensuring positive patient outcomes in the postoperative period.

While it is widely acknowledged that there are significant correlations between subjective emotional factors, such as pain and anxiety, and the occurrence of delirium [30], the precise impact of these subjective emotional factors has not been thoroughly explored to date. In previous studies, the highest incidence of delirium was observed on the day following surgery, and preoperative high anxiety only exhibited a weak positive correlation with postoperative delirium. The role of preoperative anxiety in postoperative delirium remains a topic of debate. Some studies have suggested that preoperative anxiety serves as a predictive factor for postoperative delirium [14,31,32], whereas the majority of research findings indicate no significant association between preoperative anxiety and postoperative delirium in elderly patients [12,13,14]. Consequently, further investigations focusing on the relationship between preoperative anxiety and postoperative delirium are warranted.

Previous research has established that advancing age, particularly among individuals aged over 65 years, represents a nonmodifiable risk factor for delirium [34]. This is attributed to the fact that elderly patients are more prone to conditions like malnutrition, weakness, preoperative cognitive dysfunction, and organic brain lesions (such as cerebral infarction and micro changes in white matter).

Postoperative pain, often triggered as an acute response to surgical stimulation, frequently leads to psychological, physiological, and behavioral changes in elderly patients, including anxiety, fear, depression, and sleep disturbances, often exacerbating their underlying medical conditions. Two observational studies have identified links between preoperative pain and depression and an increased risk of postoperative delirium [35,36]. Moreover, high levels of postoperative pain and the use of potent opioid medications have been associated with an elevated risk of postoperative delirium [16]. Nevertheless, our current knowledge regarding the influence of postoperative pain on postoperative delirium remains limited.

Despite numerous studies investigating the association between preoperative anxiety and postoperative pain, there is still much to learn about the underlying mechanisms of this relationship. Future research should aim to explore the neurobiological pathways through which preoperative anxiety impacts pain perception and the role of different neurotransmitters in this process. Additionally, research should explore the impact of preoperative anxiety on other outcomes, such as wound healing and infection rates, and how this may affect patients’ overall recovery. Furthermore, while various interventions have been proposed to manage preoperative anxiety, more research is needed to establish the most effective and feasible interventions for different patient populations. Future studies should also explore the impact of these interventions on long-term patient outcomes, such as quality of life and patient satisfaction. Overall, further research in this area is necessary to improve our understanding of the relationship between preoperative anxiety and postoperative pain and to develop effective interventions to manage preoperative anxiety and improve patient outcomes.

## 5. Conclusions

Enhancing our comprehension of the connections between preoperative anxiety, subsequent emotional distress, and patients’ requirements for support can potentially lead to a reduction in emotional distress among patients and enhance postoperative outcomes.

In summary, preoperative anxiety can have a profound impact on the outcome of surgical treatment. This review highlights the critical need to identify anxious patients promptly and to implement effective interventions that mitigate preoperative anxiety and its potential consequences. Healthcare professionals should proactively identify anxious patients and implement appropriate interventions, such as clinical evaluation, preoperative counseling, and management strategies.

The clinical evaluation of patients and referral for psychological intervention when necessary is crucial. Overall, addressing preoperative anxiety is an essential aspect of patient care, and healthcare professionals should take the necessary steps to address this issue and improve patient care and outcomes.

Further investigations of the connections between preoperative anxiety, perceptions of postoperative pain, and analgesic needs can be a valuable tool in improving pain management and delivering patient-centered care. Healthcare professionals can develop effective nursing interventions that aim to alleviate postoperative pain by assessing anxiety levels in the preoperative phase. Such interventions could improve the quality of care and lead to better patient outcomes. It is important to note that further research is necessary to validate these findings across various cohorts and hospital settings. Overall, exploring the associations between preoperative anxiety and postoperative pain is an essential step towards delivering compassionate and effective pain management care.

## Data Availability

No new data were created.

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
