# Peer review of "Preoperative Anxiety Impact on Anesthetic and Analgesic Use"

_medicina, 2023, doi:10.3390/medicina59122069_

Round 1

Reviewer 1 Report

Comments and Suggestions for Authors

The article deals with a very important topic, which has not received enough attention either in the literature or in clinical practice.

Although not quite novel topic, the article is written in a clear style, it is interesting and educational. More than 80 recent references are cited, which support the main authors’ claims.

Still, I do have some remarks that should be addressed.

First of all, in the Introduction section there are not enough references. In the first three paragraphs, not a single reference is cited, which I consider to be an omission and the literature, on which the points of view in the introductory part of the article are based, should be added. Furthermore, the 3rd paragraph of the introduction section describes the association of preoperative anxiety, pain and the occurrence of postoperative delirium. I don’t see the relevance of that paragraph in the context of the main topic of the manuscript so I suggest deleting the whole paragraph, or at least explaining why it should stay. In the Materials and Methods Section please state exact excluding criteria. The term “No restrictions were applied” is not enough. Also, I believe that the authors should state which population was the most prevalent in the included studies (general, surgical, geriatric or pediatric, etc.).

I believe that another paragraph (3.3) named for example “The most important consequences of perioperative anxiety” would significantly improve the quality of the present manuscript. 

Comments on the Quality of English Language

Minor editing of the English language is required.

Reviewer 2 Report

Comments and Suggestions for Authors

This paper aimed to review literatures that focused on preoperative anxiety impact on anesthetic and analgesic use. I have some comments for the authors.

1. Lines 36-39: “The prevalence of preoperative anxiety is high and carries considerable consequences for how anesthesia is administered and, ultimately, the overall outcomes of surgical procedures.” Please add relevant references here.

2. Please add relevant references for the 2nd paragraph of the introduction.

3. The authors discussed preoperative anxiety, pain, postoperative pain, and postoperative delirium in the 3rd through 5th paragraphs of the introduction. Consider revising these paragraphs to enhance the clarity of the intended ideas.

4. The authors may consider creating a dedicated section to address the impact of preoperative anxiety on anesthetic and analgesic usage, as this aligns with the primary focus of this review.

5. The authors might want to expand their discussion regarding recommendations for future studies.
